# An osteocalcin-deficient mouse strain without endocrine abnormalities

**Cassandra R. Diegel**[1], **Steven Hann**[2], **Ugur M. Ayturk**[2,3], **Jennifer C. W. Hu**[2], **Kyung-eun Lim**[4], **Casey J. Droscha**[1], **Zachary B. Madaj**[5], **Gabrielle E. Foxa**[1], **Isaac Izaguirre**[1], **VAI Vivarium and Transgenics Core**[6], **Noorulain Paracha**[7], **Bohdan Pidhaynyy**[7], **Terry L. Dowd**[8,9], **Alexander G. Robling**[4☯], **Matthew L. Warman**[2☯], **Bart O. Williams**[1☯]*

**1** Program in Skeletal Disease and Tumor Microenvironment and Center for Cancer and Cell Biology, Van Andel Institute, Grand Rapids, Michigan, United States of America, **2** Orthopedic Research Labs, Boston Children's Hospital and Department of Genetics, Harvard Medical School, Boston, Massachusetts, United States of America, **3** Musculoskeletal Integrity Program, Hospital for Special Surgery Research Institute, New York, New York, United States of America, **4** Department of Anatomy and Cell Biology, Indiana University School of Medicine, Indianapolis, Indiana, United States of America, **5** Bioinformatics and Biostatistics Core, Van Andel Institute, Grand Rapids, Michigan, United States of America, **6** Vivarium and Transgenics Core, Van Andel Institute, Grand Rapids, Michigan, United States of America, **7** Department of Biology, Brooklyn College, Brooklyn, New York, United States of America, **8** Department of Chemistry, Brooklyn College, Brooklyn, New York, United States of America, **9** Ph.D. Program in Chemistry and Ph.D. Program in Biochemistry, The Graduate Center of the City University of New York, New York, New York, United States of America

☯ These authors contributed equally to this work.
* bart.williams@vai.org

**Data Availability Statement:** The files for the RNAseq data sets reported in this manuscript have been made publicly available (BioProject ID PRJNA616424). These can be accessed at https://

## Abstract

Osteocalcin (OCN), the most abundant noncollagenous protein in the bone matrix, is reported to be a bone-derived endocrine hormone with wide-ranging effects on many aspects of physiology, including glucose metabolism and male fertility. Many of these observations were made using an OCN-deficient mouse allele (Osc⁻) in which the 2 OCN-encoding genes in mice, *Bglap* and *Bglap2*, were deleted in ES cells by homologous recombination. Here we describe mice with a new *Bglap* and *Bglap2* double-knockout (dko) allele (*Bglap/2*^p.Pro25fs17Ter) that was generated by CRISPR/Cas9-mediated gene editing. Mice homozygous for this new allele do not express full-length *Bglap* or *Bglap2* mRNA and have no immunodetectable OCN in their serum. FTIR imaging of cortical bone in these homozygous knockout animals finds alterations in the collagen maturity and carbonate to phosphate ratio in the cortical bone, compared with wild-type littermates. However, μCT and 3-point bending tests do not find differences from wild-type littermates with respect to bone mass and strength. In contrast to the previously reported OCN-deficient mice with the Osc⁻ allele, serum glucose levels and male fertility in the OCN-deficient mice with the *Bglap/2*^pPro25fs17Ter allele did not have significant differences from wild-type littermates. We cannot explain the absence of endocrine effects in mice with this new knockout allele. Possible explanations include the effects of each mutated allele on the transcription of neighboring genes, or differences in genetic background and environment. So that our findings can be confirmed and extended by other interested investigators, we are donating this new *Bglap* and *Bglap2* double-knockout strain to the Jackson Laboratories for academic distribution.

www.ncbi.nlm.nih.gov/bioproject/?term=
PRJNA616424.

**Funding:** This work was funded by the Van Andel
Research Institute and NIH grants to BOW
(AR068668), MLW and AGR (AR053237), MLW
(AR064231 and P30AR075042), and by a HHMI
Medical Research Fellowship to JCWH. The
funders had no role in study design, data collection
and analysis, decision to publish, or preparation of
the manuscript.

**Competing interests:** BOW is the recipient of a
sponsored research agreement from Janssen
Pharmaceuticals for work not directly related to
these studies. BOW also is a member of the
Scientific Advisory Board for Surrozen.

## Author summary

Cells that make and maintain bone express proteins that function either locally or systemically. The former proteins, such as type 1 collagen, affect the material properties of the skeleton, while the latter, such as fibroblast growth factor 23, enable the skeleton to communicate with other organ systems. Mutations that affect the functions of most bone-cell-expressed proteins cause diseases that have similar features in humans and other mammals such as mice, for example, brittle bone diseases for type 1 collagen mutations and hypophosphatemic rickets for mutations in fibroblast growth factor 23. Our study focuses on another bone-cell-expressed protein, osteocalcin, which has been suggested to function locally to affect bone strength <u>and</u> systemically as a hormone. Studies using osteocalcin knockout mice led other investigators to suggest endocrine roles for osteocalcin in regulating blood glucose, male fertility, muscle mass, brain development, behavior, and cognition. We therefore decided to generate a new strain of osteocalcin knockout mice that could also be used to investigate these nonskeletal effects. To our surprise, the osteocalcin knockout mice we created did not significantly differ from wild-type mice for the three phenotypes we examined: bone strength, blood glucose, and male fertility. Our data are consistent with findings from osteocalcin knockout rats but are inconsistent with data from the original osteocalcin knockout mice. Because we do not know why our new strain fails to recapitulate the phenotypes previously reported for another knockout mouse stain, we have donated our mice to a public repository so that they can be easily obtained and studied in other academic laboratories.

## Introduction

Osteocalcin (OCN) is a protein almost exclusively expressed by osteoblasts [1]. Once transcribed and translated, the 95 amino acid OCN prepromolecule is cleaved to produce a biologically active 46-amino-acid, carboxyl-terminal fragment that contains three γ-carboxyglutamic acid residues (residues 62, 66, 69 in the mouse prepromolecule) which are made post-translationally via a vitamin K–dependent process [2]. The binding of $Ca^{2+}$ to these γ-carboxyglutamic acid residues causes conformational changes that increase OCN binding to the bone mineral calcium hydroxyapatite [3]. Various states of undercarboxylated osteocalcin have been reported in bovine samples [4]; in humans, this results from limited vitamin K in the diet [5].

In humans, OCN is encoded by a single gene (*BGLAP*). In mice, OCN in encoded within a 25-kb interval on chromosome 3 that contains *Bglap* and *Bglap2*, (a.k.a., *Og1* and *Og2*), which are highly expressed in bone, and *Bglap3*, (a.k.a., *Org*), which has minimal expression in bone. The 46-residue carboxyl-terminal domains encoded by *Bglap* and *Bglap2* are 100% identical, and they differ by 4 amino acid residues from that encoded by *Bglap3*.

Mice homozygous for a large genomic deletion encompassing *Bglap* and *Bglap2* (*Osc⁻/Osc⁻*) were generated by ES-cell-mediated germline modification and described in 1996 [6]. By 6 months of age, these OCN-deficient mice had qualitatively increased cortical thickness and increased bone density relative to their control littermates. These qualitative differences were associated with significantly increased biomechanical measures of bone strength. Furthermore, OCN-deficient female mice were resistant to oophorectomy-induced bone loss. These data suggested that therapeutic reduction of OCN expression could prevent osteoporosis in humans.

Subsequent studies using $Osc^-/Osc^-$ mice broadened the biologic role for OCN by identifying wide-ranging physiologic changes when OCN is deficient. The first such report observed that $Osc^-/Osc^-$ mice had increased visceral fat and displayed elevated blood glucose associated with decreased pancreatic beta-cell proliferation and insulin resistance [7]. Subsequent work indicated OCN can enhance male fertility by inducing testosterone production and promoting germ cell survival [8]. Other observed endocrine roles for OCN include influencing fetal brain development and adult animal behavior [9], promoting adaptation of myofibers to exercise and maintaining muscle mass with ageing [10, 11], and mediating the acute stress response [12]. Cumulatively, the reports outlined above with the $Osc^-/Osc^-$ mice and, subsequently, a conditional knockout strain (Ocn-flox) made by the same investigative team [8] coupled with associated work [13–20], suggest that therapies which increase uncarboxylated osteocalcin levels may improve glucose intolerance, increase beta islet cell number, reduce insulin resistance, increase testosterone, improve male fertility, enhance muscle mass, and reduce declines in cognition.

Given the potential importance of OCN, we created new strains of OCN-deficient mice using CRISPR/Cas9 gene editing [21]. Specifically, we created mice with mutations that disrupt either *Bglap* or *Bglap2* individually or in combination. Here we report our observations regarding homozygous *Bglap* and *Bglap2* double-knockout (*Bglap/2*dko/dko) mice, which we anticipated would recapitulate the previously reported OCN-deficient phenotypes in the $Osc^-/Osc^-$ mice of increased bone mass and strength, elevated blood glucose, and decreased male fertility. We confirmed that we successfully knocked out *Bglap* and *Bglap2* by performing RNA sequencing and by measuring immunoreactive OCN in mouse serum. Consistent with previous studies, homozygous offspring with this new OCN-deficient allele (*Bglap/2*dko/dko) were born at the expected Mendelian frequency and exhibited no overt clinical phenotype. However, inconsistent with previous studies, we did not observe significantly increased bone mass or strength, significantly elevated blood glucose, significantly decreased testosterone levels, or significantly impaired male fertility.

## Results

### Generation of mice with *Bglap and Bglap2* double-knockout alleles and OCN deficiency

By simultaneously injecting Cas9 protein and guide RNAs that recognize sequences within *Bglap* and *Bglap2*, but not within *Bglap3*, we generated several founders that harbored large deletions involving *Bglap* and *Bglap2*. One founder allele, termed *p.Pro25fsTer17*, is a 6.8-kb deletion that joins exon 2 of *Bglap* to exon 4 of *Bglap2* (Fig 1A). This allele is predicted to produce a chimeric *Bglap/Bglap2* transcript with a reading frame shift after the 25th amino acid residue and a termination codon 17 residues further downstream. The predicted signal peptide encompasses 23 of these 25 amino acids. Because only two amino acids would remain after signal peptidase cleavage and because p.Pro25 is on the amino-terminal side of the biologically active domain of OCN, mice homozygous for this allele (*Bglap/2*dko/dko) should be OCN-deficient.

*Bglap/2*dko/dko offspring were born from heterozygote crosses at the expected Mendelian frequency and appeared phenotypically indistinguishable from their wild-type and carrier littermates. We confirmed that *Bglap/2*dko/dko homozygous mutants produced the frame-shifted chimeric *Bglap/Bglap2* mRNA transcript, as predicted, by RNA sequencing of freshly isolated mouse cortical bone mRNA (Fig 1B). We showed that homozygous mutants were OCN-deficient by measuring carboxylated and uncarboxylated OCN in serum (Fig 1C). Thus, the *p. Pro25fs17* allele eliminated the production of immunodetectable OCN from *Bglap* and *Bglap2*,

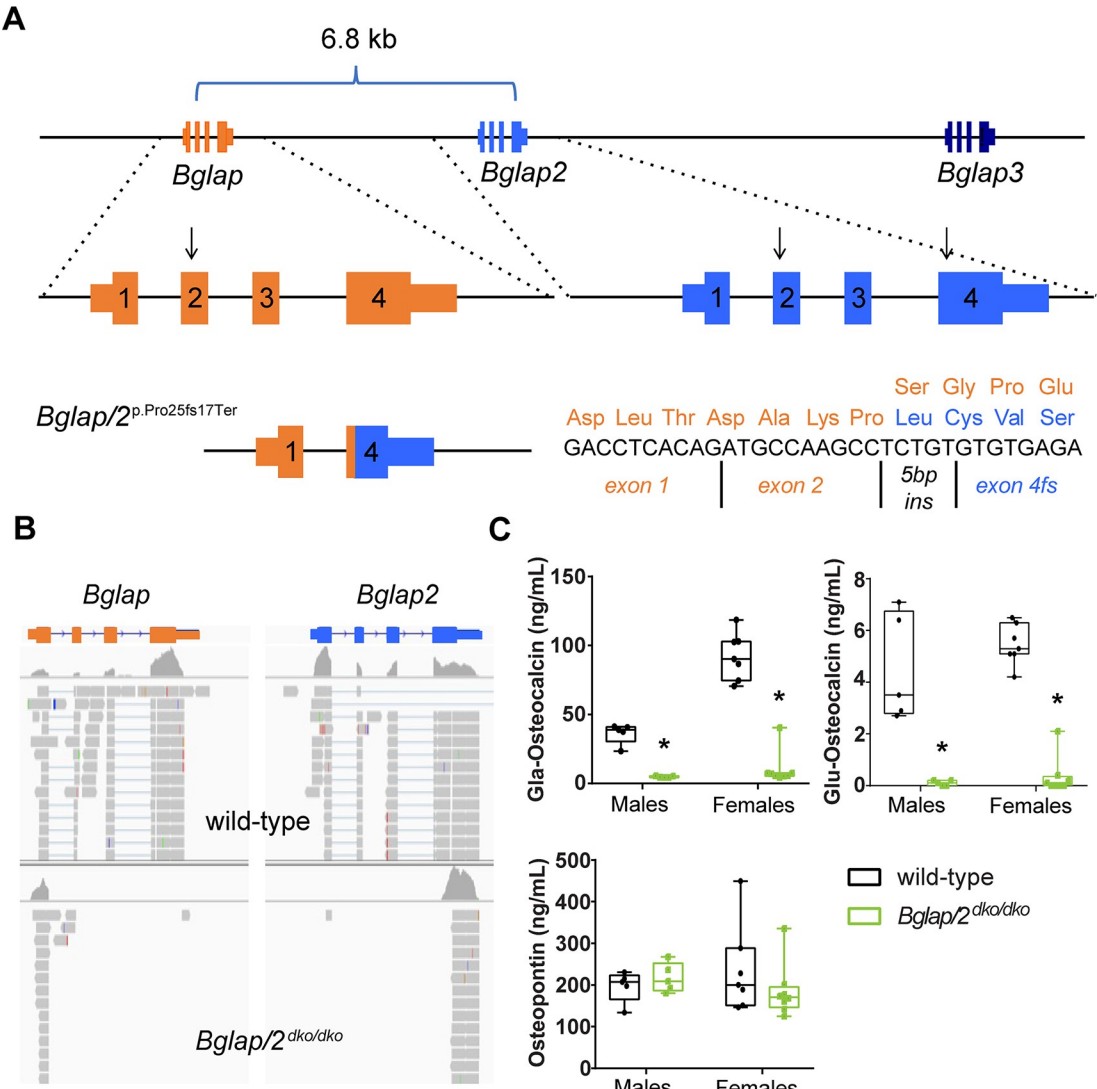

**Fig 1. Generation and validation of a *Bglap* and *Bglap2* double-knockout allele (*Bglap/2*^dko) created by CRISPR/Cas9 gene editing.** (A)Top: Schematic (not to scale) showing the 25-kb interval on mouse chromosome 3 containing *Bglap*, *Bglap2*, and *Bglap3*. The locations of the guide RNAs used to produce founders with *Bglap* and *Bglap2* intragenic or intergenic knockout alleles are indicated (arrows). Bottom, left: The *Bglap/2* double-knockout (dko) allele (*p.Pro25fs17Ter*) deletes a 6.8-kb DNA fragment extending from *Bglap* exon 2 to *Bglap2* exon 4. Bottom, right: A 5-bp insertion (5 bp ins) at the DNA ligation site creates a chimeric exon with a reading frame shift. The frame shift occurs after p.25 and terminates the protein 17 residues downstream (orange residues indicate wild-type sequence; blue indicates the first 4 frame-shifted residues). (B) Integrated Genomics Viewer screenshots of RNA sequencing data from wild-type and homozygous *Bglap/Bglap2* double-knockout mice (*Bglap/2*^dko/dko). Note the absence of sequencing reads mapping to *Bglap* exons 2, 3, and 4, and *Bglap2* exons 1, 2, and 3 in the double-knockout mice; the mapping algorithm fails to map reads to the *Bglap2* chimeric exon 2 because the chimera has a short seed length and a 5-bp insertion. (C) Serum ELISA assays for male and female 6-month-old *Bglap/2*^dko/dko (green) and their control (black) littermates for the gamma-carboxyglutamic acid (Gla) and uncarboxylated (Glu) forms of OCN as well as osteopontin (OPN). In the box and whisker plots, each box extends from the 25th to 75th percentiles, the line represents the median, and the whiskers extend to the minimum and maximum values. The following sample sizes were used: male wild-type (n = 5), male *Bglap/2*^dko/dko (n = 5), female wild-type (n = 7), and female *Bglap/2*^dko/dko (n = 8).

and, in this respect, was identical to the *Osc*⁻ allele [6] and the Cre-excised Ocn-floxed allele [8]. Because a recent report had found that loss of both OCN and osteopontin was synergistic in terms of effects on bone morphology and mechanical properties [22], we evaluated the

serum levels of osteopontin. There were no significant differences in osteopontin between *Bglap/2*$^{\text{dko/dko}}$ mice and their littermate controls (Fig 1C).

### *Bglap/2*$^{\text{dko/dko}}$ mice do not have increased bone mass or strength

Histomorphometric studies found increased bone area by 6 months of age in *Osc*$^-$/*Osc*$^-$ mice [6]. Since μCT measures have superseded histomorphometric measures for assessing bone area and bone volume, we collected femurs from 6-month-old *Bglap/2*$^{\text{dko/dko}}$ mice and compared their μCT measures to those of their wild-type littermates. We saw no significant differences in cortical or trabecular bone parameters between the *Bglap/2*$^{\text{dko/dko}}$ and wild-type mice (Fig 2A and 2B and Table 1). Femur 4-point bending assays performed on *Osc*$^-$/*Osc*$^-$ mice had revealed significantly increased bone strength [6]. We performed three-point-bending tests on 6-month-old *Bglap/2*$^{\text{dko/dko}}$ and wild-type littermates, but we did not find increased bone strength in these OCN-deficient animals (Fig 2C and S1 Table and S2 Table).

### Fourier-transform infrared imaging (FTIR) reveals increased bone crystal size and mineral maturity in *Bglap/2*$^{\text{dko/dko}}$ mice

To gain additional insight into the constitution of the bone matrix in *Bglap/2*$^{\text{dko/dko}}$ animals, we evaluated data from FTIR images collected in cortical and trabecular bone sections (Fig 3 and S3 Table). There was a significant increase ($p < 0.01$) in collagen crosslink maturity and the carbonate-to-phosphate ratio of cortical bone in the *Bglap/2*$^{\text{dko/dko}}$ mice relative to their control littermates. Carbonate can incorporate into the hydroxyapatite lattice. No significant differences were found in trabecular bone between the two groups.

### *Bglap/2*$^{\text{dko/dko}}$ mice have normal blood glucose concentration

The first reported endocrinologic role for OCN involved regulating glucose. From 1 to 6 months of age, *Osc*$^-$/*Osc*$^-$ mice consistently showed higher random blood glucose levels relative to controls ($p < 0.05$) [7]. Fasting blood glucose also increased in 6-month-old *Osc*$^-$/*Osc*$^-$ mice relative to controls ($p < 0.001$) [7, 20]. We observed no differences in blood glucose between 5- to 6-month-old *Bglap/2*$^{\text{dko/dko}}$ mice and their wild-type littermates when measured either after an overnight fast or at midday in animals with *ad libitum* access to food (Fig 4). We also observed no difference in weight between 6-month-old *Bglap/2*$^{\text{dko/dko}}$ and wild-type mice (Fig 4).

### *Bglap/2*$^{\text{dko/dko}}$ male mice have normal fertility

The second reported endocrinologic role for OCN involved male fertility. *Osc*$^-$/*Osc*$^-$ male mice had smaller testes, lower testosterone levels, and produced fewer and smaller litters than control mice [8, 19]. We therefore measured testis size, resulting litter size, and blood testosterone in 6-month-old, singly housed, virgin male *Bglap/2*$^{\text{dko/dko}}$ and wild-type littermate mice. We found no significant differences in any of these measures between the two (Fig 5), although we did note an estimated, but not statistically smaller, testicular size in the mutants (Fig 5B). We carried out an independent analysis of testosterone levels in a cohort of 10- to 15-week-old male mice. For this work, virgin male mice were singly housed for 7 d prior to blood collection from the submandibular vein. Three days later, a second blood sample was collected from each singly housed mouse and sera from both samples were sent to the P30-supported University of Virginia Ligand Assay & Analysis Core of the Center for Research in Reproduction for analysis. These analyses revealed wide variation in testosterone levels, even in the same animal

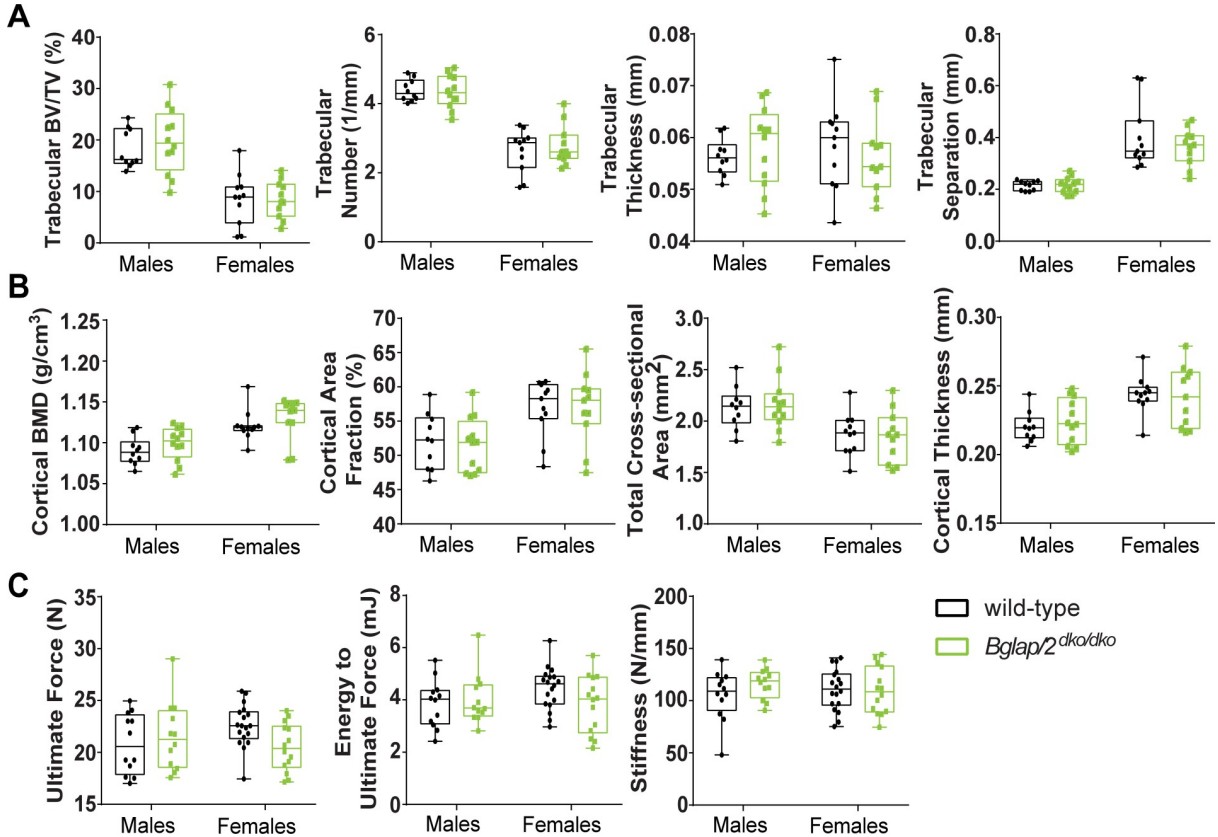

**Fig 2. OCN deficiency does not signficantly alter bone mass or strength as assessed by μCT and biomechanical testing.** In the interleaved box and whisker plots, each box extends from the 25th to 75th percentiles, the line represents the median, and the whiskers extend to the minimum and maximum values. (A) μCT analysis of trabecular bone parameters indicated no significant differences between male or female 6-month-old *Bglap/2*dko/dko (green) and their control (black) littermates. Trabecular BV/TV, trabecular number, thickness, and separation are shown. Additional measurements are included in Table 1. The following sample sizes were used for both cortical and trabecular measurements: male wild-type (n = 10), male *Bglap/2*dko/dko (n = 12), female wild-type (n = 11), and female *Bglap/2*dko/dko (n = 11). (B) μCT analysis of cortical bone parameters indicated no significant differences between male or female 6-month-old *Bglap/2*dko/dko and their control littermates. Cortical BMD, cortical area fraction, tissue area, and cortical thickness are shown as representative measurements. Additional measurements and details are included in Table 1, and sample sizes are as in panel A. (C) Biomechanical loading assessments indicated no significant differences between male or female 6-month-old *Bglap/2*dko/dko and their control littermates. Ultimate force, energy to ultimate force, and stiffness are shown as representative measurements. Additional measurements and details are included in S1 Table and S2 Table. The following sample sizes were used: male wild-type (n = 12), male *Bglap/2*dko/dko (n = 12), female wild-type (n = 18), and female *Bglap/2*dko/dko (n = 14).

taken three days apart (S1 Fig). Yet, we again did not observe the reduced levels of testosterone reported in the *Osc⁻/Osc⁻* mice [8] (Fig 5E and 5F).

## Few differences in cortical bone mRNA expression between *Bglap/2*dko/dko and control mice

RNA sequencing of cortical bone samples from *Bglap/2*dko/dko mice and their wild-type littermates confirmed the absence of wild-type *Bglap* and *Bglap2* transcripts (Fig 1B). Although we sequenced a limited number of specimens from each group, we also performed a transcriptome-wide differential expression analysis, searching for large transcriptional changes in bone tissue. We generated RNA sequencing libraries from 4-month-old male mice that on average yielded 46 million reads and 82% uniquely mapping reads. After performing *in silico* filtering to remove contaminating blood, marrow, and muscle transcripts, we identified 14 transcripts that differed significantly between the OCN-deficient and wild-type mice (after correcting for

**Table 1. μCT analysis of trabecular and cortical bone parameters for male or female 6-month-old *Bglap/2*<sup>dko/dko</sup> and their wild-type (*Bglap2*<sup>WT</sup>) littermates.** Sample sizes are shown at the top of each column.

| Index | Female | | Male | |
|---|---|---|---|---|
| | Bglap/2<sup>WT</sup> (n = 11) | Bglap/2<sup>dko/dko</sup> (n = 12) | Bglap/2<sup>WT</sup> (n = 10) | Bglap/2<sup>dko/dko</sup> (n = 12) |
| Trab.TV (mm³) | 3.181 ± 0.505 | 2.943 ± 0.469 | 3.783 ± 0.375 | 3.835 ± 0.602 |
| Trab.BV (mm³) | 0.276 ± 0.174 | 0.258 ± 0.127 | 0.693 ± 0.152 | 0.760 ± 0.249 |
| Trab.BV/TV (%) | 0.085 ± 0.048 | 0.085 ± 0.034 | 0.183 ± 0.036 | 0.198 ± 0.061 |
| Trab.Conn.D. (1/mm³) | 32.603 ± 18.917 | 33.891 ± 20.175 | 118.994 ± 22.304 | 112.240 ± 29.851 |
| Trab.SMI | 2.617 ± 0.626 | 2.542 ± 0.441 | 1.779 ± 0.406 | 1.603 ± 0.587 |
| Trab.N (1/mm) | 2.632 ± 0.588 | 2.773 ± 0.557 | 4.386 ± 0.298 | 4.350 ± 0.457 |
| Trab.Th (mm) | 0.059 ± 0.008 | 0.056 ± 0.007 | 0.056 ± 0.003 | 0.058 ± 0.007 |
| Trab.Sp (mm) | 0.401 ± 0.117 | 0.364 ± 0.067 | 0.214 ± 0.018 | 0.217 ± 0.029 |
| Trab.BMC (μgHA/cm³) | 0.261 ± 0.170 | 0.239 ± 0.116 | 0.640 ± 0.144 | 0.707 ± 0.240 |
| Cort.TV (mm²) | 6.643 ± 0.723 | 6.298 ± 0.635 | 7.059 ± 0.534 | 7.119 ± 0.741 |
| Cort.BV (mm²) | 2.910 ± 0.205 | 2.815 ± 0.188 | 2.705 ± 0.170 | 2.740 ± 0.178 |
| Cort.BV/TV (%) | 0.441 ± 0.029 | 0.450 ± 0.036 | 0.384 ± 0.020 | 0.387 ± 0.032 |
| Cort.Th (mm) | 0.244 ± 0.013 | 0.242 ± 0.021 | 0.220 ± 0.011 | 0.224 ± 0.016 |
| Cort.BMC (μgHA/cm³) | 3.264 ± 0.256 | 3.181 ± 0.221 | 2.949 ± 0.197 | 3.011 ± 0.204 |
| Cort.BMD (μgHA/cm³) | 1.121 ± 0.018 | 1.130 ± 0.025 | 1.090 ± 0.016 | 1.099 ± 0.019 |
| CAF (%) | 0.569 ± 0.039 | 0.570 ± 0.050 | 0.521 ± 0.039 | 0.515 ± 0.038 |
| T.Ar (mm) | 1.870 ± 0.193 | 1.848 ± 0.239 | 2.123 ± 0.195 | 2.166 ± 0.241 |
| Cs.Th (mm) | 0.244 ± 0.013 | 0.242 ± 0.021 | 0.220 ± 0.011 | 0.224 ± 0.016 |
| Data represents mean ± SD | | | | |

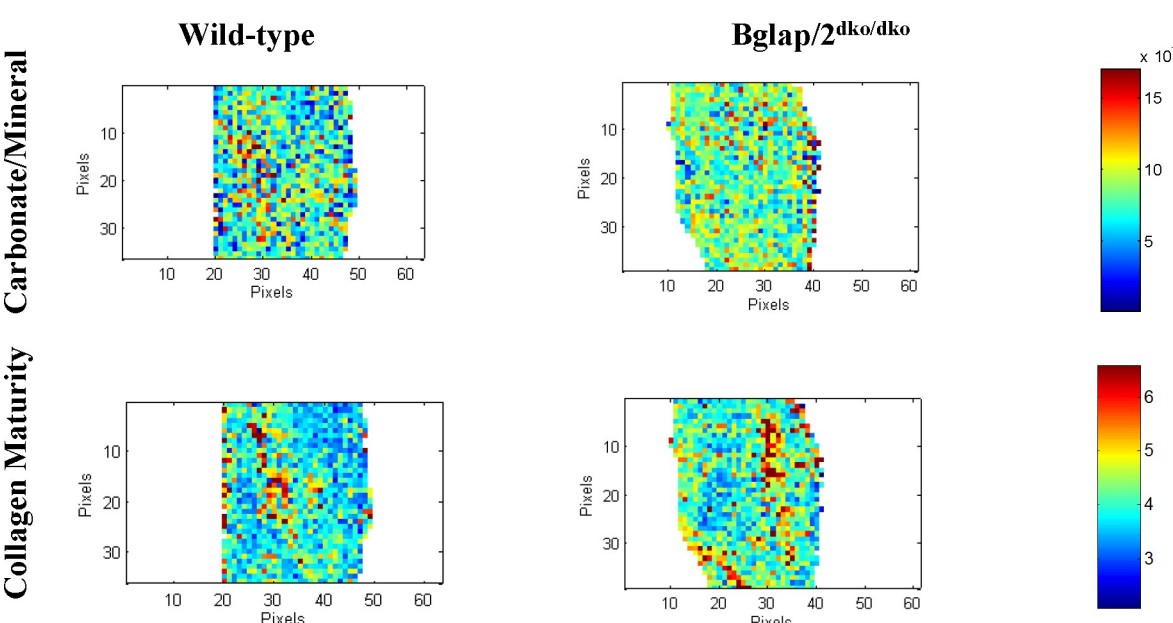

**Fig 3. FTIR showed differences in carbonate/mineral ratio and collagen maturity between *Bglap/2*<sup>dko/dko</sup> and wild-type mice.** FTIR images of cortical bone showing the spatial distribution of the variables in wild-type (n = 3) and *Bglap/2*<sup>dko/dko</sup> (n = 4) female mice. Representative images show carbonate-to-mineral ratio and collagen maturity. Additional measurements and details are included in S3 Table.

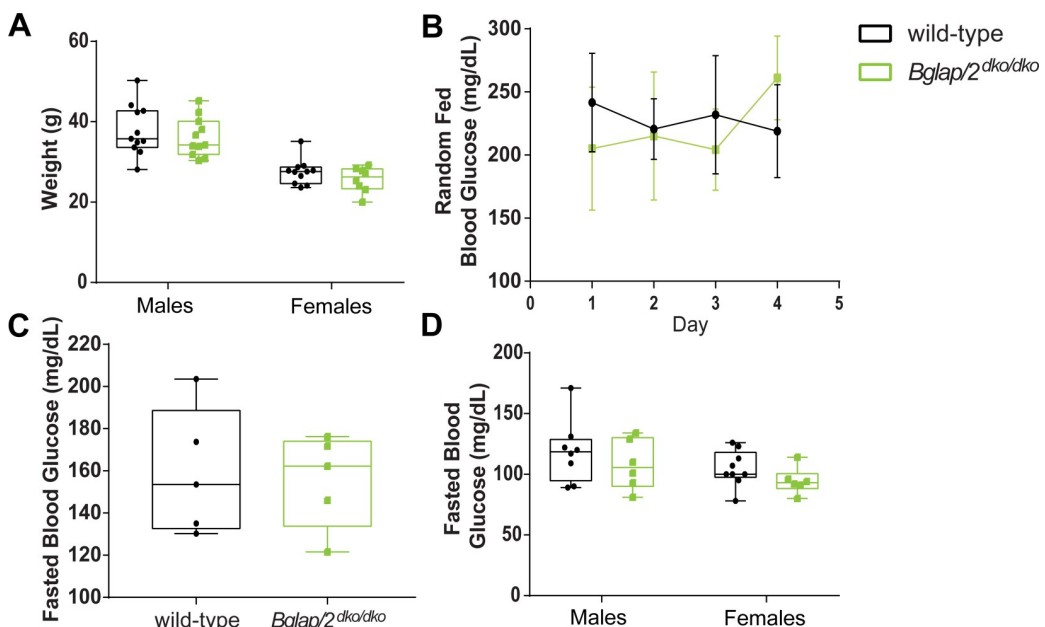

**Fig 4. No significant evidence that whole body weight, random-fed glucose levels, or fasting glucose values differ between *Bglap/2*^dko/dko^ and wild-type mice.** (A) Total weight for 6-month-old *Bglap/2*^dko/dko^ and wild-type males and females. Sample sizes were male wild-type (n = 11), male *Bglap/2*^dko/dko^ (n = 11), female wild-type (n = 11), and female *Bglap/2*^dko/dko^ (n = 8). (B) Five- to six-month-old female wild-type (n = 5) and *Bglap/2*^dko/dko^ (n = 5) mice were sampled for blood glucose concentration. Samples were taken on four consecutive days, 6 h into their light cycle while having *ad libitum* access to food. For each mouse, at least two glucose measurements were taken each day and averaged. Means and standard deviations for each genotype are shown. (C) The animals in panel 4B were then fasted for 16 h before blood glucose was again assessed. Samples were collected on two occasions approximately one week apart. At least two glucose measurements were taken on each day and averaged. The data display the means and standard deviations for each genotype. (D) After an overnight fast and at the time of euthanasia, blood glucose was measured in 6-month-old wild-type and *Bglap/2*^dko/dko^ males and females. Sample sizes are male wild-type (n = 11), male *Bglap/2*^dko/dko^ (n = 11), female wild-type (n = 11), and female *Bglap/2*^dko/dko^ (n = 8).

multiple hypothesis testing) and that had an average FPKM (fragments per kilobase of transcript per million fragments greater than 3 in mice with either or both genotypes (Table 2 and S4 Table). As expected, the mean FPKM for *Bglap* was reduced from 2157 in wild-type mice to 158 in *Bglap/2*^dko/dko^ mice. The mean FPKM for *Bglap2* did not significantly differ between wild-type and *Bglap/2*^dko/dko^ mice (1070 versus 1310 and S4 Table), with most reads in the mutant mice mapping to exon 4. Thus, despite the chimeric *Bglap/2* mRNA containing a frame-shift and premature truncation codon, it is not subject to nonsense-mediated mRNA decay. The low level of the *Bglap3* transcript (mean FPKM 10) seen in wild-type cortical bone increased 8-fold (mean FPKM 79, corrected p < 0.001) in *Bglap/2*^dko/dko^ bone is less than 2.5% of the combined FPKMs for *Bglap* and *Bglap2* in wild-type bone. It is also notable that 6 of the remaining 12 transcripts in Table 2 are encoded on chromosome 3, raising the possibility that CRISPR/Cas9 gene editing altered regulatory elements that directly affect these genes. Similar off-target mechanisms could explain the altered expression of transcripts encoded by other chromosomes. Alternatively, these gene expression changes could be the result of OCN deficiency. Although our RNA sequencing data might be underpowered to detect small changes of 2-fold or less in transcript abundance, we found no significant differences in the expression of osteoblast and osteocyte transcripts, including those of *Alpl*, *Col1a1*, *Col1a2*, *Runx2*, *Opn*, *Sost*, *Dmp1*, and *Fgf23*.

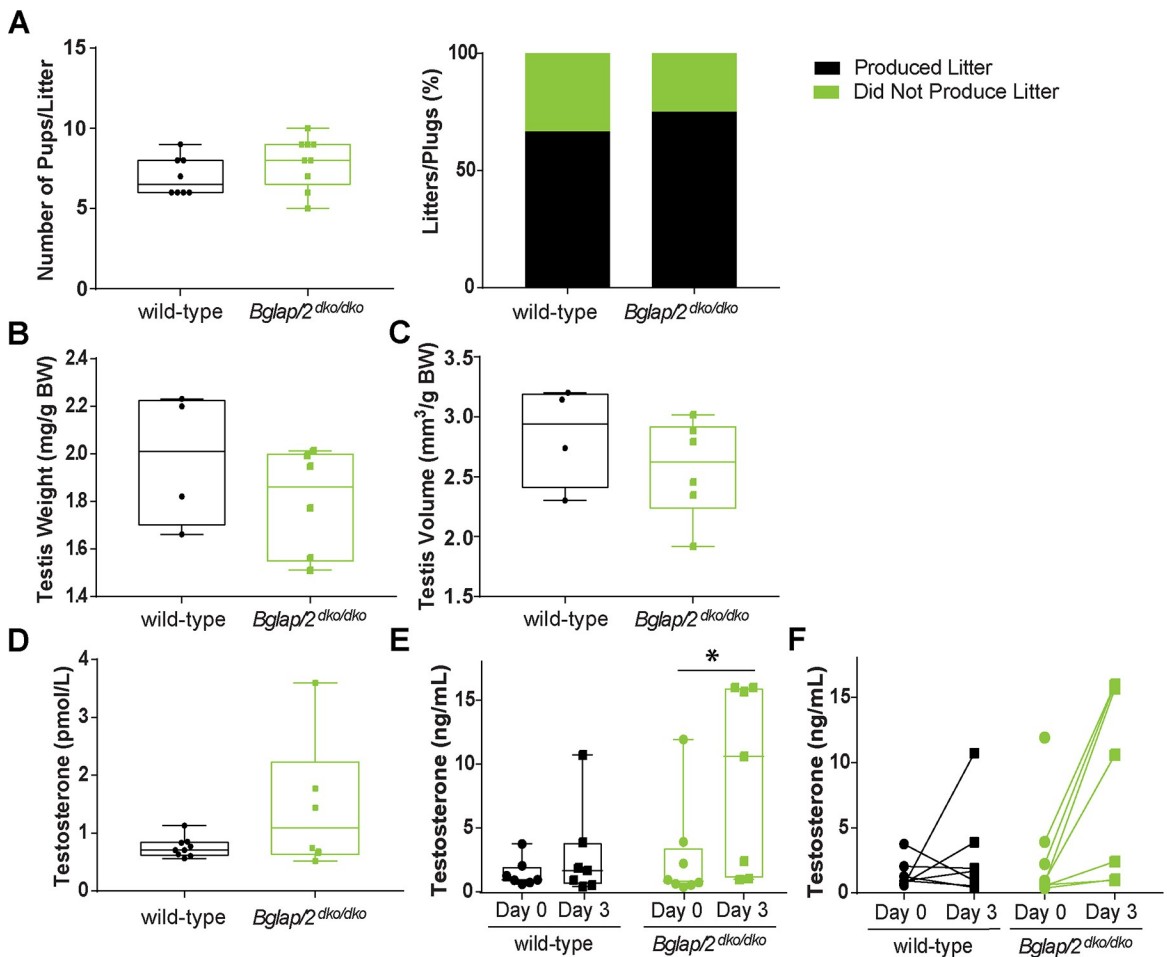

**Fig 5. No significant evidence of fertility being affected in *Bglap/2*<sup>dko/dko</sup> mice.** In the interleaved box and whisker plots, each box extends from the 25th to 75th percentiles, the line represents the median, and the whiskers extend to the minimum and maximum values. (A) The number of pups per litter and the percentage of litters per plug (n = 11 for wild-type and n = 12 for *Bglap/2*<sup>dko/dko</sup> total matings for each genotype) resulting from crosses of *Bglap/2*<sup>dko/dko</sup> males or their wild-type control littermates to control C57Bl/6J females. (B) Dry testis weight expressed as mg/g total body weight for wild-type (n = 4) and *Bglap/2*<sup>dko/dko</sup> (n = 6) males. (C) Testis volume expressed as mm<sup>3</sup>/g of total body weight for wild-type (n = 4) and *Bglap/2*<sup>dko/dko</sup> (n = 6) males. (D) Blood testosterone concentration from 6-month-old, virgin wild-type (n = 9) and littermate *Bglap/2*<sup>dko/dko</sup> males (n = 6). (E) Blood testosterone concentration (ng/mL) from 10- to 15-week-old, virgin males. Analysis of wild-type (n = 7) and *Bglap/2*<sup>dko/dko</sup> (n = 8) littermates shown at two different collection times for each animal (* represents $p > 0.05$). (F) Values from panel E plotted to show the change in testosterone of each individual from day 0 (•) to day 3 (■).

## Discussion

We created a double-knockout allele for *Bglap* and *Bglap2* (*Bglap/2*<sup>dko</sup>) and then evaluated bone properties, glucose levels, and male fertility in homozygous mutant mice and their wild-type littermate controls. We found no significant effect of OCN deficiency on bone mass and strength (Fig 2). However, using FTIR, we did find skeletal differences between *Bglap/2*<sup>dko/dko</sup> and wild-type littermates. *Bglap/2*<sup>dko/dko</sup> mice had significantly higher collagen maturity and carbonate-to-mineral ratio, suggesting osteocalcin could be involved in mineral maturation and bone remodeling. This was previously proposed for the original osteocalcin knock-out mouse [23]. The increased carbonate to mineral ratio in cortical bone is also in agreement with recent assessments of samples from the *Osc*<sup>−</sup>/*Osc*<sup>−</sup> mice maintained on a purebred C57Bl/6J background [24].

**Table 2. mRNA Transcripts with statistically significant differences in transcript abundance between *Bglap/2*<sup>dko/dko</sup> and wild-type mice.** Mean FPKMs for each cohort are shown, as are the genomic coordinates for each gene. Note that in addition to *Bglap* and *Bglap3*, 6 of the remaining 11 genes also map to chromosome 3.

| Ensembl ID | Gene ID | p-value adj | Mean dko/dko FPKM | Mean WT FPKM | Chr Start and End |
|---|---|---|---|---|---|
| ENSMUSG00000067017 | *Gm3608* | 3.22E-30 | 11.7 | 0.7 | 1; 105455971–105458829 |
| ENSMUSG00000074483 | *Bglap* | 5.38E-23 | 158.3 | 2156.7 | 3; 88383501–88384464 |
| ENSMUSG00000028081 | *Rps3a1* | 2.56E-11 | 65.2 | 287.1 | 3; 86137940–86142702 |
| ENSMUSG00000074340 | *Ovgp1* | 2.42E-06 | 4.1 | 1.0 | 3; 105973711–105987423 |
| ENSMUSG00000091383 | *Hist1h2al* | 5.55E-05 | 23.9 | 0.1 | 13; 51202727–51203065 |
| ENSMUSG00000040809 | *Chil3* | 0.000414 | 51.5 | 247.7 | 3; 106147554–106167564 |
| ENSMUSG00000074489 | *Bglap3* | 0.000477 | 77.5 | 9.8 | 3; 88368616–88372743 |
| ENSMUSG00000106508 | *4933425M03Rik* | 0.004872 | 4.0 | 0.9 | 3; 85887691–85889234 |
| ENSMUSG00000076564 | *Igkv12-46* | 0.005734 | 5.9 | 25.1 | 6; 69764523–69765022 |
| ENSMUSG00000068855 | *Hist2h2ac* | 0.009698 | 0.3 | 3.5 | 3; 96220361–96220880 |
| ENSMUSG00000051906 | *Cd209f* | 0.010412 | 3.2 | 0.8 | 8; 4102787–4105728 |
| ENSMUSG00000028104 | *Polr3gl* | 0.015468 | 50.0 | 22.5 | 3; 96577872–96594181 |
| ENSMUSG00000017861 | *Mybl2* | 0.019515 | 2.0 | 4.2 | 2; 163054687–163084688 |

In contrast to prior reports that utilized global double-knockout [6] or conditional double-knockout alleles [8], we did not observe any significant effect of OCN deficiency on blood glucose or body weight (Fig 4). Lee et al. [7] observed that OCN-deficient *Osc⁻/Osc⁻* mice had significantly increased random-fed blood glucose levels at 1, 3, and 6 months of age and significantly increased overnight-fasted blood glucose levels at 6 months of age. We powered our study to have >80% power to detect increases of a similar magnitude, but we found no difference in blood glucose between *Bglap/2*<sup>dko/dko</sup> mice and their wild-type littermates. It remains possible that the effect of osteocalcin deficiency on mouse glucose levels is much lower than previously reported, so that much larger cohort sizes or much more sensitive methods for measuring glucose utilization will be needed to identify any metabolic defect. The *Bglap/2*<sup>dko/dko</sup> mice are publicly available and can be easily obtained by an interested investigator.

Oury et al. [8] observed male *Osc⁻/Osc⁻* mice had significantly lower litter frequencies and sizes, testicular weights and volumes, and serum testosterone levels. We did not find significant effects in the *Bglap/2*<sup>dko/dko</sup> mice for these variables (Fig 5), although we did note an estimated, but not statistically significant, smaller testicular size in *Bglap/2*<sup>dko/dko</sup> mice. While we found no differences in fertility, it is possible that this is associated with more subtle perturbations in testicular development and/or function that detailed follow-up studies could evaluate. We found large variation in mouse blood testosterone in mice even when individual animals were sampled 3 d apart. This variability is consistent with previous studies and is due to mouse hepatocytes producing little sex hormone–binding globulin (SHBG), the principal carrier of testosterone in the blood, in contrast to the abundant production of SHBG in human liver [25].

We do not know why *Bglap/2*<sup>dko/dko</sup> mice did not replicate the endocrinologic phenotypes that were attributed to other *Bglap* and *Bglap2* double-knockout mice [7, 8, 19, 26]. Our RNA sequencing indicates we disrupted both the *Bglap* and *Bglap2* transcripts (Fig 1B), and our serum ELISA data indicate that OCN is undetectable in *Bglap/2*<sup>dko/dko</sup> mice. We cannot preclude the possibility that increased *Bglap3* mRNA expression compensated for the absence of *Bglap* and *Bglap2* in the *Bglap/2*<sup>dko/dko</sup> mice. Although we could not detect OCN by ELISA in *Bglap/2*<sup>dko/dko</sup> animals, the *Bglap3* protein may not be detected with this assay because it differs by 4 amino acid residues from that produced by *Bglap* and *Bglap2*. However, we think compensation by *Bglap3* is unlikely, since its cortical bone transcript abundance in the *Bglap/2*<sup>dko/dko</sup> is less than 2.5% of the cumulative *Bglap* and *Bglap2* transcript abundance in wild-type mice.

At present we do not know whether the increase in *Bglap3* mRNA expression is compensatory or an artifact of moving the *Bglap* promoter 6.8 kb nearer to *Bglap3*. The *Bglap* promoter appears to have been deleted along with *Bglap* and *Bglap2* in the *Osc⁻/Osc⁻* mice [6], but not in the *Ocn*flox conditional knockout mice [8]. Therefore, comparing RNA sequencing data across all three strains could identify allele-specific effects on gene expression that account for phenotype differences between the *Bglap/2*dko/dko mice and the mice in previous publications. Other explanations for differences in phenotype could include genetic background, vivarium environment, genetic and epigenetic changes across generations, and assay design.

Although we did not find the phenotypes previously ascribed to OCN deficiency in mice, our data do align with those reported for the OCN knockout rat. Rats, like humans, have only a single *Bglap* locus. *Bglap* knockout rats do not have elevated glucose levels, insulin resistance, or decreased male fertility [27]. To date, genetic studies in humans have not identified a role for OCN in these aspects of physiology either. No human Mendelian genetic disease has yet been attributed to loss-of-function mutations in BGLAP, in the Online Mendelian Inheritance in Man or MatchMaker exchange databases ([28, 29] https://omim.org and https://www.matchmakerexchange.org, each accessed on December 5, 2019), and there seems to be tolerance for heterozygous loss-of-function mutations at the population level (pLI = 0) [30] in the gnomAD database [https://gnomad.broadinstitute.org]. Two putative loss-of-function mutations (p.Gly27Ter19/rs1251034119 and p.Tyr52Ter/rs201282254) have carrier frequencies in the genome aggregation database (gnomAD) of 1-in-120 in African and about 1-in-150 in Ashkenazi Jewish participants, respectively. Also, one African participant and one Ashkenazi Jewish participant was homozygous for loss-of-function mutations in gnomAD. There is also no evidence from genome-wide association studies that common variants near *BGLAP* influence bone mineral density, blood glucose levels, body mass index, or risks for developing diabetes, autism, or psychiatric disease [https://www.gwascentral.org, http://pheweb.sph.umich.edu, http://www.type2diabetesgenetics.org]. Mutations in *BGLAP* do not appear to be enriched among children with autism or severe intellectual disability who have undergone research based sequencing [BioRvix: https://doi.org/10.1101/484113].

Because we did not find in the *Bglap/2*dko/dko mice the abnormalities that have been reported for the *Osc⁻/Osc⁻* mice, we chose not to perform studies interrogating the role of OCN on muscle mass, central nervous system development, behavior, or the acute stress response. Instead, are donating our mice to The Jackson Laboratory (JAX stock # 032497, allele symbol Del(3Bglap2-Bglap)1Vari). We suggest the *Osc⁻/Osc⁻* and Ocn-flox mice also be donated to a resource that facilitates public distribution, so that interested investigators can identify why different *Bglap* and *Bglap2* double-knockout alleles produce different phenotypes.

## Materials and methods

### Experimental animals

Mice were maintained in accordance with institutional animal care and use guidelines, and experimental protocols were approved by the Institutional Animal Care and Use Committee of the Van Andel Institute. Mice were housed in Thoren Maxi-Miser IVC caging systems with a 12-h/12-h light/dark cycle.

### Generation of *Bglap/Bglap2* deletion mice using CRISPR/Cas9

Alterations in the mouse *Bglap and Bglap2* alleles were created using a modified CRISPR/Cas9 protocol [31]. Briefly, two sgRNAs targeting exon 2 of OG1(AGACTCAGGGCCGCTGGGCT) and exon 4 of OG2 (GGGATCTGGGCTGGGGACTG) were designed using MIT's guide sequence generator (crispr.mit.edu.). The guide sequence was then cloned into vector

pX330-U6-Chimeric_BB-CBh-hSpCas9, which was a gift from Feng Zhang (Addgene plasmid # 42230; http://n2t.net/addgene:42230; RRID:Addgene_42230). The T7 promoter was added to the sgRNA template, and the sequence was synthetized by IDT. The PCR-amplified T7-sgRNA product was used as template for *in vitro* transcription using the MEGAshortscript T7 kit (Thermo Fisher Scientific). The injection mix consisted of Cas9 mRNA (Sigma Aldrich) (final concentration of 50 ng/μl) and sgRNA's (20 ng/μl) in injection buffer (10 mM Tris, 0.1 mM EDTA, pH 7.5) injected into the pronucleus of C57BL/6;C3H zygotes. After identifying founders, we backcrossed the line to C57BL/6 twice before intercrossing to generate animals for our study.

## Genotypic identification

To genotype the *Bglap/2*[dko] we used the following primers: OG1-E1-Fwd (ACACCATGAG GACCATCTTTC) and OG1-E4-Rev (AGGTCATAGAGACCACTCCAGC) to amplify a 517-bp wild-type product, and in a separate reaction we used OG1-E1-Fwd and OG1(E2)-OG2(E4)-Rev (AAGCTCACACACAGAGGCTTGG) to amplify a 238-bp knockout product. These animals are available from Jackson Laboratories (stock number 032497).

## Blood chemistry analysis

Blood was harvested into an EDTA-treated tube at the time of euthanasia via intracardiac puncture, spun down at 8,000 rpm for 6 min, and plasma collected and stored at –80˚C until use. Enzyme immunoassays were used to measure plasma concentrations of mouse Glu-osteocalcin (MK129;Takara), mouse Gla-osteocalcin (MK127; Takara), mouse osteopontin (MOST00; R&D Systems), and mouse testosterone (55-TESMS-E01; Alpco), according to the manufacturers' recommendations. Statistical analysis was performed used a linear regression model to test for differences between genotypes via R. We also collected blood (see below) from 10- to 15-week-old males for testosterone measurements analyzed independently by the P30-supported University of Virginia Ligand Assay and Analysis Core Laboratory.

## Collection of samples for skeletal analysis

Femurs were collected at 26 weeks of age; the right hindlimb was fixed in neutral buffered formalin for μCT analysis, and the left hindlimb was frozen in saline-saturated gauze for biomechanical testing.

## μCT and mechanical testing

Formalin-fixed femora were scanned, reconstructed, and analyzed as previously described [32]. Briefly, 10 μm resolution, 50-kV peak tube potential, 151-ms integration time, and 180˚ projection area were used to collect scans on a Scanco μCT-35 desktop tomographer. The distal 60% of each femur was scanned, thresholded, and reconstructed to the 3[rd] dimension using Scanco software. Standard parameters related to cancellous and cortical bone mass, geometry, and architecture were measured [33].

For evaluation of bone mechanical properties, frozen femur samples were brought to room temperature over a 4-h period, then mounted across the lower supports (8 mm span) of a 3-point bending platen and mounted in a TestResources R100 small force testing machine. The samples were tested in monotonic bending to failure using a crosshead speed of 0.05 mm/ s. Parameters related to whole bone strength were measured from force/displacement curves as previously described [34, 35].

## Fourier-transform infrared imaging (FTIR) analysis

Femora from female, 6-month-old wild-type and *Bglap/2*<sup>dko/dko</sup> mice were cleaned of soft tissue, processed, and embedded in polymethylmethacrylate (PMMA) [36]. Longitudinal sections, 2 μm thick, were mounted on infrared windows where spectral images were collected at a 4 cm$^{-1}$ spectral resolution and approximately 7 μm spatial resolution from a Spotlight 400 Imaging system (Perkin Elmer Instruments, Shelton, CT USA). Background spectra were collected under identical conditions from clear $Ba_2F$ windows and subtracted from sample data by instrumental software. IR spectra were collected from three areas (approximately 500 μm x 500 μm) of cortical bone per sample. The spectra were baseline-corrected, normalized to the PMMA peak at 1728 cm$^-$, and the spectral contribution of PMMA embedding media was subtracted using ISYS Chemical Imaging Software (Malvern, Worcestershire, UK). Spectroscopic parameters of carbonate-to-mineral ratio, crystallinity, mineral-to-matrix ratio, collagen maturity, and acid phosphate were calculated. The carbonate-to-mineral ratio is the integrated area ratio of the carbonate peak (850–890)/$\nu_1 \nu_3$ $PO_4$ band (900–1200 cm$^{-1}$), while the mineral-to-(collagen)-matrix ratio is the integrated area ratio of the $\nu_1 \nu_3$ $PO_4$ band (900–1200 cm$^{-1}$) / amide I band (1590–1712 cm$^{-1}$). The mineral crystallinity parameter corresponds to the crystallite size and perfection as determined by X-ray diffraction and is calculated from the intensity ratios of subbands at 1030 cm$^{-1}$ (stoichiometric apatite) and 1020 cm$^{-1}$ (nonstoichiometric apatite). The collagen maturity parameter is the ratio of nonreducible (mature) to reducible (immature) collagen cross-links, which is expressed as the intensity ratio of 1660 cm$^{-1}$/1690 cm$^{-1}$. The acid phosphate content in the mineral is measured from the peak height ratio of 1128/1096. The result for each parameter was reported as a histogram, describing the pixel distribution in the image. The mean value of the distribution was reported and associated color-coded images were generated at the same time by ISYS. Data for each measured parameter are expressed as mean ± standard error of the mean for each group. The data for all measured parameters were found to be normally distributed as analyzed by the Shapiro-Wilk tests did not find enough evidence to conclude any measured parameters were nonnormally distributed. The average values were compared by the Student's independent *t*-test for significant differences between groups. Differences for each measured parameter were considered statistically significant when p $\leq$ 0.05.

## Baseline and random glucose measurements

Cohorts of 5- to 6-month-old female mice, wild-type (n = 5) and *Bglap/2*<sup>dko/dko</sup> (n = 5), were subjected to four random glucose measurements taken 6 h into their light cycle by tail nick using a glucose meter (AlphaTRAK; Zoetis) on four consecutive days. These same cohorts had glucose measurements obtained by tail nick after an overnight fast during which the animals had access to water on two occasions one week apart.

Additional cohorts of mice were fasted overnight, but with access to water, prior to euthanasia by $CO_2$ inhalation. Glucose levels were measured immediately after euthanasia. Animals were euthanized by $CO_2$ inhalation and glucose levels were immediately measured from blood collected by tail nick using a glucose meter.

Glucose data were analyzed using a linear mixed-effects model via the R package *lme4* [37] to account for repeated sampling. Normality of the residuals was verified visually using a qq-plot.

## Measurement of testis size and weight

Testes were removed from 6-month-old males after euthanasia and fixation. Fatty tissue was carefully removed from each testis and the sample dried prior to weighing on an analytical

scale to determine dry weight. Testis weights was normalized to body weight (mg/g BW). Each testis was imaged and the length and width calculated using Nikon's NIS-Elements documentation software. Testis volume was normalized to body weight (mm$^3$/g BW). Statistical analysis was performed used a linear regression model to test for differences between genotypes via R.

## Assessment of fertility

Male mice of mating age (7–16 weeks of age) were singly housed for 1 week before mating. Males were mated to C57BL/6J females in the evening and vaginal plugs checked the following morning. Number of pups per litter were compared between genotypes, and the percentage of vaginal plugs resulting in delivery and the number of pups per delivery were noted. Logistic mixed-effects regression with a random intercept for paternal mouse was used to determine if the rate of conception differed significantly between the two genotypes. A Poisson mixed-effects model with a random intercept for paternal mouse was used to determine if the number of pups per delivery differed significantly between the two genotypes. Both models were analyzed via the R v 3.5.2 (https://cran.r-project.org/) package *lme4* [37].

Methods for testosterone measurement are provided in the previous section on blood chemistry analysis. We performed two independent testosterone measurements. One cohort was collected from 6-month-old virgin males that were singly housed for 3 d prior to sample collection. Plasma was isolated from whole blood and used to perform the mouse testosterone ELISA (ALPCO) in-house following the manufacturer's protocol. The second cohort of 10- to 15-week-old virgin males were singly housed for 7 d prior to blood collection from the submandibular vein. Three days later, we collected a second sample from each male using the contralateral submandibular vein. The samples were collected in the morning each day and mice were sampled in the same order. Single cages were removed from the housing unit and the mouse removed for sampling. Serum was isolated from whole blood and sent to the University of Virginia Ligand Assay and Analysis Core to measure testosterone values independently.

Statistical analysis was performed in R v3.6.0 using a linear mixed-effects model with random intercepts for each mouse and litter. Linear contrasts with a Benjamini-Hochberg adjustment for multiple testing were used to test specific hypotheses.

## Cortical bone RNA sequencing and data analysis

Tibial cortical bone was recovered from 4-month-old male wild-type (n = 3) and *Bglap/2*$^{\text{dko/dko}}$ (n = 5) mice immediately following euthanasia by $CO_2$ inhalation as previously described [38]. Samples were frozen in liquid nitrogen, pulverized, and suspended in TRIZol. Total RNA was recovered using the PureLink™ RNA Mini Kit (Invitrogen) according to the manufacturer's instructions, including on-column DNAse digestion (PureLink™ DNase Set, Invitrogen). RNA quality was assessed using a Bioanalyzer (2100 Bioanalyzer, Agilent). RNA abundance was quantified based on the height of the 28S ribosomal peak since co-purifying contaminants confounded RNA abundance determinations based on RINs.

Twenty-five ng of total RNA was used to construct an RNA-seq library for each sample. RNA was converted to cDNA using the SMART-Seq v4 Ultra Low Input RNA kit. The cDNA was fragmented using a sonication device (Covaris, E200). Library construction was completed using the ThruPLEX DNA-seq Kit (Rubicon Genomics). Size selection was performed with AMPure XP Beads (Beckmann Coulter) as per the manufacturer's directions. Quality and mean fragment size of library samples were assessed with the Bioanalyzer prior to sequencing. Libraries were sequenced on a NextSeq 550 deep sequencing unit (150 cycles, paired-end, Illumina) at the Biopolymers Facility at Harvard Medical School, Boston, MA. Reads were mapped using STAR aligner to the *Mus musculus* genome (mm10, Ensembl release 89) [39].

Sequencing and alignment quality was analyzed with FastQC [40], RSeQC [41], and Picard (broadinstitute.github.io/picard). Read counts were calculated using the Subread package.

Because blood, bone marrow, and muscle could not be completely removed from the tibial cortical bone prior to RNA extraction, we computationally removed reads representing 894 transcripts which we had previously shown to comprise about 10% of all cortical bone mRNA and come from these contaminating tissues [35]. We then used EdgeR [42] to measure transcript abundance by calculating fragments per kilobase of transcript per million fragments mapped (FPKM).

We quantified -fold changes in expression between *Bglap/2*<sup>dko/dko</sup> and wild-type transcripts using the DESeq2 package [43]. Reported in Table 2 are those transcripts having a mean FPKM greater than 3 in wild-type, *Bglap/2*<sup>dko/dko</sup>, or in both mice, and those that changed significantly (adjusted p values < 0.05) after correcting for multiple hypothesis testing using the Benjamini-Hochberg method. S4 Table includes adjusted p values for all transcripts regardless of FPKM.

## Supporting information

**S1 Fig. Mean blood testosterone difference between day 0 and day 3 of 10- to 15-week-old, virgin males.** The mean difference of the change in testosterone is displayed using a +/- 95% confidence interval (CI). The difference in testosterone was calculated by subtracting the day 0 value from the day 3 value ($^*$ represents $p > 0.05$).
(TIF)

**S1 Table. Biomechanical testing on male *Bglap/2*<sup>dko/dko</sup> and age-matched wild-type animals.** Individual measurements for each mouse are shown for ultimate force, stiffness, and energy to FU. *Bglap/2*<sup>dko/dko</sup> (KO/KO, n = 12) and wild-type (WT, n = 12) are noted, and the average and standard deviation are presented for each variable.
(DOCX)

**S2 Table. Biomechanical testing on female *Bglap/2*<sup>dko/dko</sup> and age-matched wild-type animals.** Individual measurements for each mouse are shown for ultimate force, stiffness, and energy to FU. *Bglap/2*<sup>dko/dko</sup> (KO/KO, n = 14) and wild-type (WT, n = 18) are noted, and the average and standard deviation are presented for each variable.
(DOCX)

**S3 Table. FTIR imaging quantitation for cortical and trabecular bone.** Quantitation of variables for *Bglap/2*<sup>dko/dko</sup> (KO/KO, n = 4 or 3) and wild-type (n = 3) female mice are shown.
(DOCX)

**S4 Table. Differential expression calculated using the DESeq2 package of cortical bone mRNA transcripts between male *Bglap/2*<sup>dko/dko</sup> and male wild-type littermate mice.** This table includes adjusted p values (controlling for multiple hypothesis testing) for all transcripts regardless of FPKM.
(XLS)

## Acknowledgments

We thank other members of the Williams, Warman, Robling, and Dowd Laboratories for suggestions and technical assistance, and Dr. Nicholas Stylopoulos for advice regarding serial glucose measurements. Key members of the VARI Vivarium and Transgenics Core include Bryn Eagleson, Adam Rapp, Nicholas Getz, Audra Guikema, Tristan Kempston, Tina Schumaker, and Malista Powers. We thank David Nadziejka for editorial assistance.

## Author Contributions

**Conceptualization:** Cassandra R. Diegel, Casey J. Droscha, Matthew L. Warman, Bart O. Williams.

**Data curation:** Cassandra R. Diegel, Steven Hann, Ugur M. Ayturk, Gabrielle E. Foxa, Alexander G. Robling, Matthew L. Warman, Bart O. Williams.

**Formal analysis:** Cassandra R. Diegel, Steven Hann, Ugur M. Ayturk, Jennifer C. W. Hu, Kyung-eun Lim, Zachary B. Madaj, Gabrielle E. Foxa, Isaac Izaguirre, Noorulain Paracha, Bohdan Pidhaynyy, Terry L. Dowd, Alexander G. Robling, Matthew L. Warman, Bart O. Williams.

**Funding acquisition:** Matthew L. Warman, Bart O. Williams.

**Investigation:** Steven Hann, Jennifer C. W. Hu, Noorulain Paracha, Terry L. Dowd, Alexander G. Robling, Matthew L. Warman, Bart O. Williams.

**Methodology:** Cassandra R. Diegel, Steven Hann, Ugur M. Ayturk, Jennifer C. W. Hu, Kyung-eun Lim, Zachary B. Madaj, Isaac Izaguirre, VAI Vivarium and Transgenics Core, Bohdan Pidhaynyy, Terry L. Dowd, Alexander G. Robling, Matthew L. Warman, Bart O. Williams.

**Project administration:** Cassandra R. Diegel, Matthew L. Warman, Bart O. Williams.

**Resources:** Matthew L. Warman, Bart O. Williams.

**Supervision:** Cassandra R. Diegel, Alexander G. Robling, Matthew L. Warman, Bart O. Williams.

**Validation:** Cassandra R. Diegel, Steven Hann.

**Writing – original draft:** Ugur M. Ayturk, Zachary B. Madaj, Terry L. Dowd, Matthew L. Warman, Bart O. Williams.

**Writing – review & editing:** Cassandra R. Diegel, Gabrielle E. Foxa, Alexander G. Robling, Matthew L. Warman, Bart O. Williams.

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
