## [Decision Letter · Decision Letter 0]

1 Oct 2019

Dear Dr Williams,

Thank you very much for submitting your Research Article entitled 'An Osteocalcin-deficient mouse strain without endocrine abnormalities' to PLOS Genetics.

The manuscript was fully evaluated at the editorial level and by independent peer reviewers. As you will see, all three reviewers are supportive of the importance of the question and the central findings that the new mouse model of osteocalcin deficiency challenges early findings on another osteocalcin mouse model. Overall, we agree that the work has the potential to provide new information important for bone and mineral metabolism research.

At the same time, there are some significant issues that were raised by the reviewers, as well as a number of minor points. The first relates to the statistical significance of some of the results and question of whether or not the numbers of animals that were used are sufficient to support the validity of the claims. Given the importance of the findings in re-interpreting earlier data in the field, it is particularly important that the findings are robust and supported by adequate numbers of experimental mice.

The second major recommendation is that in addition to fasting glucose levels, provocative testing such as glucose tolerance should be carried out to further evaluate the claim that osteocalcin deficiency does not lead to endocrine abnormalities.

We therefore ask you to modify the manuscript according to the review recommendations before we can consider your manuscript for acceptance. Your revisions should address the specific points made by each reviewer.

[LINK]

Yours sincerely,

John F Bateman

Associate Editor

PLOS Genetics

Gregory Barsh

Editor-in-Chief

PLOS Genetics

Reviewer's Responses to Questions

**Comments to the Authors:**

Reviewer #1: During the last twenty three years, the bone and mineral metabolism research community (and the NIH institutes and numerous other national and international funding agencies that support it) have devoted a great deal of intellectual energy and resources on some very provocative, albeit controversial, ideas about osteocalcin (OCN) – a 46 amino-acid protein that is made in bone and binds avidly to the hydroxyapatite mineral. OCN is produced and secreted almost exclusively by osteoblasts, terminally differentiated cells responsible for the synthesis and mineralization of the bone matrix during the development of the skeleton and its periodic regeneration throughout life. Osteoblasts are short-lived cells originating from mesenchymal progenitors that are replaced depending on the demand for bone formation in a particular location and time. OCN secreted by osteoblasts contains three γ-carboxyglutamic acid residues that are removed by the acidic pH created during the resorption of bone by osteoclasts.

According to ideas originated and propagated by Gerard Karsenty and co-workers, osteoblasts comprise an endocrine organ and decarboxylated OCN is a hormone. The circulating levels of this “new hormone” are, therefore, dependent on the rate of bone turnover, a.k.a. remodeling. Specifically, Karsenty and collaborators have claimed in a series of high profile publications over a period of several years that OCN acts on multiple organs and tissues including bone, pancreas, liver, adipose, muscles, testicles, and the brain to regulates functions ranging from bone mass accumulation, to body weight, adipocity, glucose metabolism, energy utilization, male fertility, mentation, and behavior. Notably, all these claims have been based on the results of studies of a single mouse model with genetic deletion of OCN, generated in the Karsenty laboratory over 23 years ago, but never made available to other investigators seeking to reproduce the results.

However, unlike hormones produced and released by dedicated cells in response to external stimuli, the number of osteoblasts and thereby the circulating levels of osteocalcin inexorably change throughout life as a result of physiologic or pathologic changes of bone itself that can be acute or chronic, systemic or localized, and reversible or irreversible. Examples are skeletal development, growth, adaptation of the skeleton to mechanical forces, fracture healing, changing calcium needs, stress, menstrual cycle, pregnancy, lactation, menopause, aging, hyper- or hypo-parathyroidism, hyperthyroidism, hypercortisolemia, Paget’s disease, bone tumors, etc. Furthermore, medications – approved after extensive trials with thousands of subjects and used by millions for the treatment of osteoporosis – decrease or increase serum osteocalcin levels without any effect on glucose homeostasis, testosterone production, muscles, or behavior. The glaring shortcomings and incongruence of the Karsenty ideas with physiology, pathophysiology, clinical medicine, and pharmacology notwithstanding, the concerns with the hypothesis that bone and osteocalcin regulate many other tissues have progressively intensified over the years by the fact that other groups have failed to reproduce the observations of Karsenty and colleagues in different mice or in a rat model with OCN deletion. Still major journal review articles and prestigious teaching textbooks have continued until now to publish the Karsenty claims as proven biologic facts.

In the study of Diegel et al, the authors have generated their own mouse model of loss of OCN function using CRISWPR/Cas9-mediated gene editing. Homozygous mice with deletion of the two alleles that encode OCN have undetectable mRNA for either allele in bone and no circulating OCN. Additionally, RNA sequencing of cortical bone samples from the OCN deficient mice show minimal differences from the wild type control mice. Unlike the Karsenty model, these mice have normal bone mass and strength, as well as normal blood glucose and male fertility. They do, however, exhibit increased bone crystal size and maturation of hydroxyapatite, consistent with earlier evidence and the general consensus that OCN plays a role in mineralization. The authors conclude that their OCN-deficient mouse has no endocrine abnormalities. The methods of analysis of the phenotype are state of the art, the results are solid and totally convincing, and the manuscript is concisely and clearly written.

Minor point

Page 3 of the Introduction, second line form the bottom: Spell out FPKM.

Reviewer #2: The authors used Crispr-CAS technology to generate a mouse in which a large portion of both of the osteocalcin genes in mice were deleted. They showed that the mRNA generated from the deleted genome encodes a truncated protein likely to be non-functional and showed that antisera detected no immunoreactive osteocalcin in KO mouse blood. MicroCT showed no difference in bone mass between KO and WT and bone strength, measured by 4-point bending assays was normal in the KO. They noted tha the KO had an increased crystallinity and decrease in acid phosphate in the KO. Fasting blood glucose was normal in 6-month-old KO, as was weight. Fasting and random blood sugar levels were similar in KO and WT mice. Fertility, testis size and testosterone levels were normal in KO males. Thus, the knockout mouse failed to reproduce the reported abnormalities in the bone.

1. While the Karsenty group did see changes in fasting blood glucose between WT and KO mice, they showed much more dramatic differences when mice were challenged with a glucose tolerance test. I think it’s important for the current investigators to challenge their mice with a glucose tolerance test to see if that brings out a difference in glucose metabolism missed by the studies that they performed.

2. Testis weight and volume measurements involved only four WT mice. The weights and volumes of the KOs were lower, but apparently not statistically so. Particularly given the wide variation in the WT testis weight (20-30%), the authors need to do a power calculation to determine how many mice need to be studied to avoid missing a real difference, and repeat this study with adequate power. Testosterone levels vary substantially in male mice, depending on whether they are being housed with females or not. No mention was made of precautions to minimize variation among the mice. Was testosterone measurement made at the same time of day to minimize variation due to diurnal variation? When the findings are that the mice are not different, then substantial efforts should be made to avoid missing real differences.

3. The nature of the deletion in the KO mouse appears to allow the possible expression of a peptide fragment of osteocalcin. How do the authors know that this isn’t sufficient to rescue the pancreatic and muscle effects of osteocalcin? This seems unlikely to me, but needs to be discussed more fully by the authors. Has there been any structure-function analysis of these proposed actions of osteocalcin?

Reviewer #3: This is a concise, well-constructed, and articulate study of novel Bglap1/2 KO mice, designed to characterise the physiological role of osteocalcin, which has been the subject of much research since the original publication of a KO mouse describing an endocrine phenotype for the bone protein KO mouse. Herein these studies confirm a role for osteocalcin in bone matrix, though not bone mass and strength. Crucially, there are no changes in plasma glucose or male fertility, which is at odds with a significant body of previous work.

As such, this work represents a valuable addition to the literature, and I commend it. I have no specific comments regarding the studies performed; suffice to say that they are generally appropriate, well executed, and clearly presented.

I have a few minor comments that the authors might wish to consider:

1. Please clarify more clearly the statistical analyses performed; it is not clear from the figure legends or methods for all of the experimental studies.

2. Related to (1), given the magnitude of the previously observed effects with Blglap KO mice (from the lab of Karsenty e.g. change in fasting plasma glucose), please confirm that the ‘n’ numbers used for the studies herein are sufficiently powered to detect such effects, given the variability in any given dataset? Also, are the values (e.g. fasting plasma glucose, bone density) for the WT mice in this study in line with previous work?

3. Please clarify in more detail the background of the mice, and specifically any differences between this new transgenic and the previous (for example, what is reported with respect to Bglap3 for previous mice lines?).

4. At times in the Results, it is not clear whether the authors are referring to their own data, or that of previous studies (quoting stats from previous papers makes this more confusing still). Please just check the syntax in places.

**Have all data underlying the figures and results presented in the manuscript been provided?**

Reviewer #1: Yes

Reviewer #2: Yes

Reviewer #3: Yes

PLOS authors have the option to publish the peer review history of their article (what does this mean?). If published, this will include your full peer review and any attached files.

Reviewer #1: Yes: Stavros Manolagas, MD, PhD

Reviewer #2: No

Reviewer #3: No

---

## [Editor Report · Decision Letter 1]

2 Mar 2020

Dear Dr Williams,

We are pleased to inform you that your manuscript entitled "An Osteocalcin-deficient mouse strain without endocrine abnormalities" has been editorially accepted for publication in PLOS Genetics. Congratulations!

Yours sincerely,

John F Bateman

Associate Editor

PLOS Genetics

Gregory Barsh

Editor-in-Chief

PLOS Genetics

Comments from the reviewers (if applicable):

**Data Deposition**

http://datadryad.org/submit?journalID=pgenetics&manu=PGENETICS-D-19-01327R1

**Press Queries**

---

## [Editor Report · Acceptance letter]

29 Apr 2020

PGENETICS-D-19-01327R1 

An Osteocalcin-deficient mouse strain without endocrine abnormalities 

Dear Dr Williams, 

We are pleased to inform you that your manuscript entitled "An Osteocalcin-deficient mouse strain without endocrine abnormalities" has been formally accepted for publication in PLOS Genetics! Your manuscript is now with our production department and you will be notified of the publication date in due course.

With kind regards,

J&J Graphics

PLOS Genetics

On behalf of:
